# Role of Sterilization on In Situ Gel-Forming Polymer Stability

**DOI:** 10.3390/polym16202943

**Published:** 2024-10-21

**Authors:** Elena O. Bakhrushina, Alina M. Afonina, Iosif B. Mikhel, Natalia B. Demina, Olga N. Plakhotnaya, Anastasiya V. Belyatskaya, Ivan I. Krasnyuk, Ivan I. Krasnyuk

**Affiliations:** A.P. Nelyubin Institute of Pharmacy, I.M. Sechenov First Moscow State Medical University (Sechenov University), Moscow 119048, Russia; bakhrushina_e_o@staff.sechenov.ru (E.O.B.); aafonina205@gmail.com (A.M.A.); demina_n_b@staff.sechenov.ru (N.B.D.); plakhotnaya_o_n@staff.sechenov.ru (O.N.P.); belyatskaya_a_v@staff.sechenov.ru (A.V.B.); krasnyuk_i_i_1@staff.sechenov.ru (I.I.K.J.); krasnyuk_i_i@staff.sechenov.ru (I.I.K.)

**Keywords:** in situ systems, sterilization, smart polymers, polymer stability, thermoreversible polymers, ion-sensitive polymers, polymer protection

## Abstract

In recent years, stimulus-sensitive drug delivery systems have been developed for parenteral administration as a depot system. In situ systems incorporate smart polymers that undergo a phase transition at the site of administration. All parenteral and ocular dosage forms must meet sterility requirements. Careful selection of the sterilization method is required for any type of stimuli-sensitive system. Current sterilization methods are capable of altering the conformation of polymers or APIs to a certain extent, ultimately causing the loss of pharmacological and technological properties of the drug. Unfortunately, the issues of risk assessment and resolution regarding the sterilization of stimuli-sensitive systems, along with ways to stabilize such compositions, are insufficiently described in the scientific literature to date. This review provides recommendations and approaches, formulated on the basis of published experimental data, that allow the effective management of risks arising during the development of in situ systems requiring sterility.

## 1. Introduction

Gel-forming in situ systems (ISSs) are drug delivery systems that undergo a phase transition at the site of administration under the influence of various physical, physicochemical or chemical factors. The most commonly used compositions respond to three main stimuli: changes in temperature, pH and ionic composition [1,2]. Many authors additionally distinguish systems capable of stimulus-sensitive gel formation by phase inversion (diffusion of solvent into surrounding soft tissue), moisture activation and photosensitivity, along with those formed during redox reactions and in the presence of enzymes [3]. These systems have many advantages over conventional dosage forms and can be administered orally, buccally, rectally, vaginally, cervically, parenterally (including intratumoral administration) and intranasally. Ophthalmic ISSs are also of particular interest [1,2,4].

Depending on the composition, in situ gels are highly mucoadhesive, providing a long contact time of the drug on the mucous membrane, a slow controlled release of the active pharmaceutical ingredient (API) at the site of administration and a prolonged therapeutic effect, which reduces the frequency of dosing and improves patient compliance. The increased bioavailability of the drug allows lower doses to be used, reducing the risk of side effects. Injectable ISSs require lower manufacturing costs and are more convenient to use than traditional depot systems such as films and implants. The advantage of ophthalmic ISSs over traditional eye drops and gels is their ability to provide accurate dosing and promote precorneal retention [1,4,5,6,7].

Injectable and ophthalmic ISSs, along with drugs in contact with wounds, according to pharmacopoeial requirements, must meet the criteria for drug sterilization. The purpose of sterilization is to prevent microbial contamination of the drug product, which may result in the degradation of the API during storage and the development of infection in the patient’s body [8,9,10,11].

Special attention must be paid to the sterilization process of gel-forming ISSs, as an inappropriate method may result in the destruction of the polymer and the drug, loss of activity and alteration of the release profile [10]. Unfortunately, to date, the scientific literature has not adequately addressed the issues of risk assessment and risk mitigation in the sterilization of stimulus-sensitive systems or ways to stabilize such compositions. Despite the existence of a number of studies, the data obtained by different scientific groups are not harmonized and compiled, which makes it difficult to build effective methodologies for the pharmaceutical development of sterile ISSs.

The aim of this review is to evaluate the effect of sterilization on the stability of ISSs based on different stimuli and polymers and to identify effective approaches for the in situ stabilization of compositions requiring sterilization.

## 2. Current Sterilization Methods and Risks for the ISS

The sterility of a drug product can be ensured by terminal sterilization or by creating aseptic conditions. This requires well-trained staff, a high level of supervision and extreme care to avoid subsequent contamination. Pharmaceutical guidelines recommend the use of terminal sterilization, as aseptic processing does not provide the same level of safety as terminal sterilization [8,11,12,13].

Terminal sterilization methods include physical, chemical and physicochemical methods. Physical methods include thermal, radiation and sterile filtration. Chemical sterilization is performed by gases (ethylene oxide, ozone, etc.) and liquids (alcohols, aldehydes, surfactant solutions) and is used rather rarely [8,11].

Thermal methods include sterilization with saturated steam under pressure (autoclaving) and hot air (air sterilization), which require temporary exposure to high temperatures (for autoclaving, 15 min at 121–124 °C; for air sterilization, 30 min at ~180 °C). These conditions cause the death of microorganisms as a result of the irreversible denaturation of proteins. Autoclaving is a highly efficient method that is suitable for large-scale production, but it can lead to the decomposition of the API and polymer and change the physicochemical properties of the ISS [8,9,10,11].

Gamma radiation sterilization is a simple, convenient and effective method due to its high penetration and low temperature capability. In addition, this method allows continuous operation with the accurate measurement of the absorbed radiation dose. As with autoclaving, it can have a severe impact on in situ gels [5,8,9,12].

Membrane filtration is a safe method based on the physical removal of the microorganisms’ using a membrane with a pore size of 0.22 µm and does not require high temperatures or radiation. However, this method is less reliable and is only used when other sterilization methods cannot be used [9,11,13].

Table 1 provides an overview of the main advantages highlighted by researchers when performing the described sterilization methods on ISSs.

To effectively manage the identified risks, an individual selection of the sterilization method is required, depending on the nature and properties of the polymer composition. The strict control of critical parameters before and after sterilization to maintain consistent product quality may also be recommended. Suggestions for these approaches are described below. The main risks for most commonly used sterilization methods are described in Figure 1.

## 3. Basic Polymers for ISS Development

As mentioned above, three main groups of polymers are distinguished according to their gelling mechanisms: thermosensitive, pH-sensitive and ion-sensitive [1,14,15,16,17,18,19,20,21].

### 3.1. Thermoreversible Polymers

These polymers are single-phase solvation-like systems in aqueous media at temperatures below the lower critical solution temperature (LCST). Above the LCST, the solvated polymer chains dehydrate and hydrophobic interactions between the polymer chains increase, leading to a sol–gel transition. These systems are reversible and can return to the sol state at temperatures below the LCST [1].

Examples of thermosensitive polymers include poloxamers, also called pluronics (BASF, Germany), xyloglucan, cellulose derivatives (methylcellulose (MC), hydroxyethyl cellulose (HEC), hydroxypropyl methylcellulose (HPMC), sodium carboxymethylcellulose (NaCMC)) and the chitosan/beta-glycerophosphate system [1,18,22].

#### 3.1.1. Chitosan

The most popular polymer contained in in situ systems is chitosan [23]. Chitosan is a natural cationic copolymer consisting of D-glucosamine and N-acetyl-D-glucosamine linked by β-1,4-glycosidic bonds. It is obtained by the deacetylation of chitin isolated from the exoskeleton of crustaceans, insects or fungi [21,23,24,25]. Gel formation occurs in the alkaline medium as a result of deionization, the decrease in apparent density of the polymer, hydrophobic interactions and the formation of hydrogen bonds (pH sensitive polymer). In combination with the weak base beta-glycerophosphate, it forms a thermoreversive gel due to ionic interactions [26].

Chitosan has good mucoadhesiveness due to the presence of free hydroxyl and amino groups that allow the polymer to form hydrogen and electrostatic bonds with mucin [24]. Due to its natural origin, chitosan material varies widely in molecular weight, degree of deacetylation and purity [24].

Chitosan degradation occurs upon heating and is accompanied by a decrease in viscosity and loss of molecular weight (Mw) [24]. Chitosan films degrade when autoclaved at 121 °C for 15–30 min. However, the stability of chitosan solutions has been less studied [25].

In a study by Jarry et al., 2% chitosan solutions were autoclaved at 121 °C. After sterilization, a decrease in Mw, loss of dynamic viscosity, gelation rate and mechanical strength of chitosan glycerophosphate gels were found, while the ability to gel in situ was retained [25].

Chitosan is also sensitive to gamma irradiation. This sterilization method causes significant breaks in the backbone, leading to a decrease in Mw [24].

The irradiation of highly deacetylated chitosan fibers and films at doses up to 25 kGy caused cleavage of the main chain, accompanied by a decrease in Mw and glass transition temperature. Irradiation under anoxic conditions did not cause changes in functional groups and was characterized by higher Mw values, in contrast with gamma irradiation in air [27].

The effect of gamma irradiation on chitosan solutions, in contrast with the solid forms of the polymer, has been less studied. In the study by Jarry et al., the gamma irradiation of a chitosan solution resulted in a significant decrease in viscosity, which the authors attributed to polymer degradation [28].

#### 3.1.2. Block Co-Polymers of Polyethylene Oxide and Polypropylene Oxide (Poloxamers)

Poloxamers are the second most popular polymers used in ISSs [23]. They are triblock co-polymers of poly(ethylene oxide)-b-poly(propylene oxide)-b-poly(ethylene oxide) (PEO-PPO-PEO). Poloxamers are amphiphilic and form micelles in water in which hydrophobic domains of PPO interacting through van der Waals forces form a hydrophobic core, and hydrophilic domains of PEO occupy a hydrophilic shell and form hydrogen bonds with water molecules [10,29]. Poloxamer 407 contains 70 wt.% PEO and 30 wt.% PPO and has a gelation temperature of 35 °C [6,18].

The mechanism of gelation involves polymer desolvation and micelle aggregation, leading to a decrease in fluidity of the system [6,29,30]. The gelation temperature depends on the polymer concentration and the hydrophobicity of the molecule, i.e., the more hydrophobic it is, the lower the phase transition temperature [18]. The polymer concentration increase leads to a decrease in gelation temperature [14].

Aqueous solutions of poloxamers are stable in the presence of acids, bases and metal ions. Studies have been performed on the stability of poloxamer-based gels after sterilization and the data correlate with each other. For example, in a study by Burak et al., poloxamer-based gels 407 were sterilized by autoclaving at 121 °C for 20 min or at 105 °C for 30 min. After sterilization, a decrease in viscosity, an increase in gelation temperature and a decrease in temperature coefficient were observed. These results were due to the polymer degradation that poloxamers undergo when exposed to heat in an oxygen-containing environment. In this case, the samples sterilized at 121 °C underwent more changes than those sterilized at 105 °C. Nevertheless, the gels retained their pseudoplastic properties after autoclaving and the authors of the study concluded that autoclaving at 105 °C can be used for terminal sterilization [18].

Summarizing the above data, we can say something about the relative thermal stability of poloxamers, but it should be noted that the degradation of poloxamers under the influence of high temperatures in an oxygen-containing medium is possible [18].

The stability of poloxamers after sterilization by gamma irradiation was investigated in a study by Nesseem et al. In situ samples of sparfloxacin gel based on poloxamer 407 and poloxamer 188 were exposed to gamma irradiation at 20 °C. The study of IR spectra of the drugs before and after sterilization showed that none of the components were affected by gamma irradiation [31].

In a study by Saher et al., an in situ hydrogel combination of poloxamer 407 (24%) and poloxamer 188 (15%) with levofloxacin hemihydrate was exposed to gamma irradiation at two doses—high (25 kGy) and low (0.6 kGy). Sterilization caused significant changes in both physical properties and drug content, even at low irradiation doses [32]. However, the researchers also noted that it was likely that these changes were due to the degradation of the drug rather than the polymer [32].

In another study, mono-hydrogels of poloxamer 407 at concentrations of 10, 15 and 25% were exposed to gamma irradiation at different doses (15, 25, 50, 75, 100 kGy). An increase in the viscosity of the gels was observed, associated with the cross-linking of the polymer chains, but the samples retained non-Newtonian properties [33].

The data presented demonstrated the relative stability of poloxamers with respect to gamma irradiation.

#### 3.1.3. Cellulose Derivatives

Cellulose is a linear polysaccharide consisting of β (1 → 4) linked d-glucose molecules [21]. Cellulose derivatives have a high phase transition temperature, which can be reduced by physical and chemical modification. The addition of sodium chloride reduces the gelation temperature of MC from 45–50 °C to 33–34 °C, and the reduction of the phase transition temperature of HPMC from 70–90 °C to 40 °C is achieved by reducing the molar substitution with hydroxypropyl [21].

The mechanism of gel formation of these polymers is the dehydration of the polymer with increasing temperature, which leads to the enhancement of hydrophobic interactions between HPMC molecules, polymer–polymer association and the formation of a network structure [34].

HPMC is unstable to gamma radiation. Upon irradiation at a dose of 18 kGy, a rupture of the pyranose ring and cleavage of the beta-glycoside bond to form an aldehyde group was observed. Chain cleavage leads to a decrease in viscosity of aqueous solutions that is proportional to the irradiation dose. The sterilization of extended-release matrix solid dosage forms containing diltiazem and HPMC resulted in a significant decrease in release time, suggesting a less effective control of drug release [35]. These results are in agreement with another study where the irradiation of dry HPMC powders at doses of 0–25 kGy was characterized by a proportional decrease in viscosity; with increasing irradiation doses, the solutions became Newtonian liquids [36].

CMC is also unstable to sterilization by gamma irradiation. In a study by Adams et al., CMC gels exposed to gamma irradiation lost their gel structure and became Newtonian liquids with reduced viscosity [37].

In another study, irradiation of 3 and 5% CMC solutions at doses of 0–30 kGy resulted in a decrease in viscosity that was proportional to the irradiation dose. However, the degree of degradation was lower for the 5% solution, which may be due to the fact that higher concentrations interfere with the diffusion and interaction of radicals with polymers and promote chain cross-linking [38].

Studies on the stability of CMC during autoclaving have yielded conflicting results. In a study by Zhukovskii et al., autoclaving of the CMC solution resulted in a decrease in viscosity. The viscosity loss increased with an increasing CMC concentration, reaching 76% for a 6% solution [39].

The difference in published data on this subject is probably related to the quantitative content of hydroperoxides in the initial raw material.

For example, researchers [40,41] found that hydroxypropyl cellulose (HPC) feedstock contains a mixture of hydroperoxides (HPO), including organic hydroperoxides (ROOH) and hydrogen peroxide (H_2_O_2_). The amount of HPO varies from batch to batch and from manufacturer to manufacturer. The H_2_O_2_ impurity may be related to its use in pulp processing to reduce the average Mw. The formation of organic hydroperoxides appears to be related to the oxidation processes of cellulose derivatives, which can occur under the action of chemical and physical factors.

It should be noted that the presence of hydroperoxides is also possible in other stimuli-sensitive polymers. In one study, hydroperoxide levels were determined in poloxamers 188, 338 and 407, but their content was negligible [42]. Hydrogen peroxide can also be used to produce low molecular weight chitosan, alginate and guar gum [43,44,45,46,47]. It is important to control the content of HPO in the raw materials before sterilization, otherwise the loss of gel viscosity due to the degradation of polymers under the action of hydroperoxide free radicals is possible.

During autoclaving, hydroperoxides and transition metal ions initiate the formation of free radicals that cause the cleavage of glycosidic bonds and loss of gel viscosity. For example, in a study by Ji et al., autoclaving of HPMC gel from five different batches resulted in viscosity loss in all five samples, but to different degrees [41]. In another study, HPMC hydrogel was autoclaved and a decrease in the viscosity, hardness and adhesiveness of the formulation was observed [48]. In a study [49], the autoclaving of 2 and 3% HPMC gels resulted in a slight decrease in viscosity.

Thus, cellulose derivatives may undergo oxidative degradation during autoclaving due to the presence of hydroperoxides in the raw materials, but the degree of degradation is variable and depends on the impurity content.

#### 3.1.4. Xyloglucan

Xyloglucan (Figure 2) is a polysaccharide of plant origin based on b-1,4-D-glucose linkages partially substituted by a-(1,6)-linked xylose units, some of which are substituted by b-D-galactose at O-2 or by the galactose/fucose disaccharide. After xyloglucan is treated with fungal b-galactosidase, which removes more than 35% of the galactose residues, the polymer becomes thermosensitive [17].

In the study [17], xyloglucan hydrogels were sterilized by autoclaving and gamma irradiation at room temperature and in dry ice. The samples showed good stability and almost no change in their properties after autoclaving and gamma irradiation in dry ice. However, sterilization by gamma irradiation at room temperature resulted in a reduction of Mw to 75%, preventing thermogel formation due to the cleavage of the glucan chain and side chains [17]. These results suggest the need to protect xyloglucan under gamma irradiation sterilization conditions by using dry ice.

### 3.2. Ion-Sensitive Polymers

Ion-sensitive polymers have anionic groups that complex with monovalent (Na^+^) and/or divalent (Mg^2+^ and Ca^2+^) cations present in physiological fluids and form hydrogen bonds with water, resulting in the formation of double helix binding zones and a three-dimensional gel network [1]. The phase transition is influenced by the polymer concentration, temperature and the nature and concentration of the cations [50].

Examples of ion-sensitive polymers include pectin, sodium alginate and gums (gellan gum, xanthan gum, guar gum) [1,23].

#### 3.2.1. Gums

Gellan gum is an anionic polysaccharide from the bacterium Sphingomonas elodea consisting of the repeating tetrasaccharide element glucuronic acid, rhamnose and glucose at a 1:1:2 ratio [1,21,50].

Guar gum is a heteropolysaccharide extracted from guar beans consisting of repeating linkages of galactose and mannose [23].

Xanthan is an anionic polysaccharide derived from the bacterium Xanthomonas campestris [23]. The main chain consists of 1–4 linked ß-D-glucopyranose residues, the side chains are represented by a trisaccharide consisting of glucuronic acid and two mannose residues attached to the O-3 atom of every other glucose residue. Half of the terminal mannose residues contain a pyruvate group [51,52]. In an aqueous solution, it exists in the form of a double helix [53].

Xanthan is sensitive to high temperatures. In a study by Bindal et al., autoclaving solutions of the polymer resulted in a change from the double helix conformation to a filamentary conformation with subsequent hydrolysis, as indicated by a decrease in Mw. These changes were inversely proportional to the concentration of xanthan, which is associated with active intermolecular interactions and the entanglement of molecules at high concentrations [53].

Similar results were obtained in another study on the sterilization of xanthan gum hydrogel [48].

A study [54] investigated the effect of sterilization on the stability of various polymers. According to the results, autoclaving guar gum at a concentration of 0.5% or higher did not cause any changes in apparent viscosity, while the sterilization of xanthan gum caused a significant decrease in apparent viscosity and desirable shear liquefaction rheology [54].

In another study, the in situ autoclaving of guar gum-based gel resulted in a slight decrease in viscosity, indicating its resistance to heating [55].

The gamma irradiation of xanthan gum solutions resulted in a decrease in Mw and apparent viscosity associated with the cleavage of glycosidic linkages. The decrease in Mw was proportional to the increase in irradiation dose. A comparison of Fourier-transform infrared spectroscopy (FTIR) spectra showed that the basic structure of the polysaccharide chain was preserved, but the absorption peak in the UV spectrum at 265 nm indicated the formation of new carbonyl groups [51].

In a study by Hayrabolulu et al., aqueous solutions of xanthan gum at different concentrations (0.5–4%) were exposed to gamma irradiation at doses of 2.5–50 kGy and a decrease in Mw was observed. The degradation rate was higher at lower concentrations. This was due to the increased mobility of (-OH) due to the reduced viscosity of the solution. At low concentrations, the polymer chains are further apart, which prevents the recombination of macro radicals [52]. The above studies suggest that xanthan gum is unstable to gamma irradiation.

Guar gum is also unstable to gamma irradiation. The irradiation of samples with a moisture content of 10% at a dose of 5 kGy at room temperature resulted in a significant increase in the lowest gelation concentration, decrease in pH, viscosity of dispersions, average Mw and changes in color and rheological properties. These changes were due to the cleavage of the polymer chain with the release of galacturonic acid and the oxidation of functional groups, as shown by FTIR analysis [56].

In another study, the gamma irradiation of solid samples of guar gum resulted in a decrease in Mw, intrinsic viscosity and shear viscosity, but no changes in molecular structure other than chain shortening were observed [57].

A number of independent studies [58,59,60,61] have demonstrated the stability of gellan gum-based ISSs under sterilization by autoclaving and gamma irradiation. No changes in pH, gelling ability and viscosity were observed. The IR spectrum showed the absence of chemical reactions during sterilization.

Thus, it can be concluded that gellan gum is the most stable stimuli-sensitive polymer from the gum group and can be recommended for use in the formulation of dosage forms requiring sterilization.

#### 3.2.2. Alginate

Alginate is a natural polysaccharide composed of R-1-guluronate (G) and beta-D-mannuronate (M) monomers linked by 1–4 glycosidic linkages [21], with homopolymeric regions of M and G and regions approximating a disaccharide repeating structure (MG). Gel formation occurs when sodium alginate interacts with calcium ions [21].

Sodium alginate exhibits instability upon heating. In Stoppel et al., the autoclaving of hydrogel resulted in a significant change in mechanical properties [62]. In another study, the exposure of alginate hydrogel to steam also caused a dramatic decrease in viscosity, which was attributed to a decrease in the Mw of the polymer due to degradation [14]. In Alipour et al., the autoclaving of sodium alginate gel (2–4%) also caused a significant decrease in viscosity, which was attributed to the hydrolytic degradation of the polymer [49].

Alginate has also been shown to be unstable when sterilized by gamma irradiation. When alginate solution was irradiated, a decrease in Mw from 300,000 to 25,000 was observed at a dose of 100 kGy, with viscosity decreasing almost to a minimum at 10 kGy [63].

In a study [64], the effect of sterilization by gamma irradiation on the stability of alginate films was investigated. It was found that low doses of irradiation (0.1–0.5 kGy) resulted in an increase in the strength of the films, which was associated with the cross-linking of the polymer chains, but, as the irradiation dose was further increased, the strength began to decrease. This may be due to the breaking of glycosidic bonds under the action of free radicals [64].

### 3.3. pH-Sensitive Polymers

pH-sensitive polymers form a gel at a pH below the pKas (basic materials) or at a pH above the pKas (acid materials). Polymers with weak acidic properties deprotonate at an alkaline pH, forming a negative charge that leads to the electrostatic repulsion of polymer molecules and the formation of viscous solutions. The gelation process is influenced by factors such as Mw, an ionic strength of the solution [1]. Examples of pH sensitive polymers are chitosan, carbomers and polycarbophil [1,22].

Chitosan, as mentioned above, forms a gel in an alkaline medium as a result of deionization, a decrease in the apparent density of the polymer, hydrophobic interactions and hydrogen bond formation. The stability of chitosan during sterilization has been discussed above.

#### 3.3.1. Acrylic Acid Derivatives (Carbomers)

Carbomers (trade name Carbopol) are synthetic anionic polymers consisting of acrylic acid cross-linked to sucrose by allyl ethers or pentaerythrityl allyl ethers. The carboxyl groups of acrylic acid are capable of accepting and releasing protons as the pH changes. Carbopol forms a gel at a pH above a pK of about 5.5 as a result of ionic interactions between negative charges in the polymer and cations in the environment [14,21]. Carbopol has high mucoadhesiveness, and the interaction with mucin is realized by four mechanisms: hydrogen bonding, electrostatic interaction, hydrophobic interaction and mutual diffusion. The disadvantage of Carbopol is its high acidity; therefore, it is used in combination with other polymers [21,22].

Carbopol remains stable when autoclaved. The sterilization of 1% Carbopol solutions by autoclaving caused a slight decrease in viscosity, but the samples retained pseudoplastic properties [37].

In another study, autoclaving Carbomer 940P at concentrations ranging from 0.05% to 0.10% caused a slight increase in apparent viscosity [54]. The increase in viscosity may be due to the cross-linking of the polymer chains.

Autoclaving of the Carbopol 940 hydrogel did not cause any significant changes in the analyzed composition. Although the pH of the hydrogel was significantly decreased after sterilization, the composition returned to its original value after three weeks [14].

These results are in agreement with another study where the autoclaving of Carbopol gel (0.25–1%) caused only a slight loss of viscosity, except for the 0.25% solution where the viscosity was halved. This may be due to the greater hydration of the polymer at low concentrations, making it more susceptible to hydrolysis.

Studies on the stability of Carbopol after sterilization by gamma irradiation have shown mixed results. In the work of Adams et al., gamma irradiation (30 kGy) of 1% Carbopol solutions caused a significant decrease in the viscosity of the formulations [37]. In another study, the 25 kGy gamma irradiation of a 2.5% carbomer gel did not cause any pH changes, the observed viscosity decrease was not significant and the gel was suitable for use [65].

The differences in the results of the studies may be due to the different irradiation doses used and the influence of other antioxidant components of the gel (EDTA, sodium metabisulfite). EDTA chelates metal ions, which can initiate the formation of free radicals [40].

#### 3.3.2. Polycarbophil

Polycarbophil (Figure 3) is a polymer of polyacrylic acid cross-linked with divinyl glycol that forms a gel when the pH is raised above its pKa of about 6.0 ± 0.5 [66].

The effect of sterilization on the stability of polycarbophil polymers has been studied little. In one study [48], autoclaving a 2% polycarbophil gel had a negligible effect on its viscosity and mechanical properties. In another study, polycarbophil gel was autoclaved, but its stability was not investigated [67].

#### 3.3.3. Zeolites

One of the latest achievements of world science is the consideration of zeolites as promising stimuli-sensitive polymers (Figure 4).

The zeolite imidazole framework (ZIF), consisting of metal ions (Zn, Co, Ni, Cu and others) surrounded by tetrahedrons of nitrogen atoms and linked by the imidazole cycle is used as a targeted delivery system.

ZIFs are capable of degradation at a low pH [68], which accounts for their frequent use for the delivery of chemotherapeutic agents, since many types of tumors are characterized by acidosis. A systematic review by J. Hao et al., 2021, mentioned the therapeutic use of ZIFs as delivery systems for doxorubicin, 5-fluorouracil, curcumin, cisplatin and miR-34a [69].

Although, according to PubMed, there are no published studies on the problems of the sterilization of zeolites by various methods, it can be assumed that these polymers, like other mineral compounds, are relatively stable when exposed to high temperatures. In the process of obtaining artificial zeolites, autoclaving at temperatures up to 200 °C is used [70]. For zeolites and zeolite-like imidazole frameworks, the intrinsic antibacterial effect is actively studied and is already used to sterilize drugs containing this excipient [71]. Therefore, the study of stability issues during the sterilization of this group of polymers is obviously not relevant and will not be considered in this review.

### 3.4. Polymer Mixtures with Different Gelation Stimuli

It is important to note that the ISS usually contains several polymers with different stimuli. This approach makes it possible to improve rheological and gelling properties and to reduce the concentrations of polymers used [7,50,72]. At the same time, polymers can influence each other and change stability (both increase and decrease stability), which is important to consider when choosing a sterilization method.

For example, in a study by Asasutjarit et al., an in situ gel of diclofenac sodium based on poloxamer 407 (20%) in combination with poloxamer 188 (11%) and Carbopol 940 (0.1%) was autoclaved at 121 °C and 15 p.s.i. for 20 min. Sterilization did not significantly affect the rheological properties or the gelation temperature of the samples analyzed. Changes in pH and drug content were caused by the chemical degradation of diclofenac sodium [10].

Another study was performed on temperature-sensitive in situ gels of fluconazole based on poloxamer 407 (20%) with the addition of Tween 80, benzalkonium chloride and Carbopol 934 (0.1%) in a borate buffer solution. After autoclaving, the sol–gel transition temperature, fluidity and behavior of the test samples did not change [73].

In Shastri et al., in situ gels of poloxamer 407 (15%) with Carbopol (0.4%), poloxamer 407 (15%) with sodium alginate (0.7% and 0.9%) and poloxamer (15%) with xanthan gum (0.5%) were autoclaved at 121 °C and 15 p.s.i. for 20 min. Sterilization had a negligible effect on viscosity, pH and drug content [72].

Different results were obtained in another study [74]. The gel based on poloxamer 407 (2.5%) and alginate (2%) was characterized by a marked decrease in complex viscosity after sterilization by autoclaving, and the elastic modulus could not be determined. Both polymers used in the work, alginate and poloxamer, interacted strongly with each other and the interaction was modified by the autoclaving process [74].

In the Gupta et al. study, a gel based on poloxamer 407 (9%) combined with chitosan (0.25%) was sterilized under the same conditions. Autoclaving did not affect the pH, gelling ability or viscosity of the formulation. After gelation under the influence of high temperatures, the system reverted to a liquid after cooling [6].

A study by Gupta et al. [66] showed that autoclaving had no effect on the viscosity or pH of the in situ gel of poloxamer 407 (16%) with polycarbophil (0.3%).

In another study, gels based on poloxamer 407 (25%) and its combination (25%) with MC (5%) and (25%) with HPMC (3%) were autoclaved at 121 °C for 30 min. Sterilization was shown to have no effect on the viscosity of the formulations [75].

In a study by Venkatesh et al., an in situ gel containing poloxamer 407 (20%) and HPMC (1.5%) was subjected to gamma irradiation at a dose of 15.76 kGy at 25 °C. No significant changes in gel formation temperature and time, spray time or drug release were observed [5].

Gamma irradiation of in situ gels of poloxamer 407 (20%) and Carbopol 934 (1%) also did not cause significant changes in temperature and gelation time, application time or drug release and was also shown to be effective as a sterilization method in a sterility test [76].

The present studies show that the addition of poloxamer increased the stability of chitosan, alginate and xanthan gum, which are sensitive to autoclaving, along with HPMC and Carbopol, which are degraded by gamma irradiation.

Poloxamer micelles are capable of binding to another polymer, such as chitosan, through hydrophobic interactions. Such bonds provide greater gel strength [77]. A similar mechanism is presented for the combination of poloxamer with alginate, where the polysaccharide chains form hydrogen bonds with the micelles, acting as a cross-linking agent [74]. It appears that these processes are responsible for the stability of combinations of poloxamer with other polymers. Hydrophobic interactions and hydrogen bonds between micelles and polymer chains provide gel stabilization during sterilization because additional energy is required to break these bonds. Polymer aggregation can also make it difficult for polymers to interact with water, thus protecting against hydrolysis.

The loss of viscosity in the study [74] may have occurred due to the use of low concentrations of poloxamer, which were insufficient to protect the alginate from hydrolytic cleavage.

In addition, when sterilized by gamma irradiation, high concentrations of poloxamer reduced the mobility of free radicals and promoted chain cross-linking, thus maintaining the viscosity value.

A stability study of ISSs of ciprofloxacin hydrochloride based on Carbopol 934 and HPMC after sterilization by gamma irradiation showed the stability of the formulation. The IR spectra of the drugs before and after sterilization showed similar peaks and no changes in appearance were observed [34].

In the study by Srividya et al., an in situ gel of ofloxacin based on Carbopol (0.5%) and HPMC (1.5%) was subjected to autoclaving. No changes in pH, gelation capacity or viscosity were observed. Turbidity caused by HPMC precipitation at an elevated temperature disappeared after cooling [78].

In a study [79], sterilization by the in situ autoclaving of gels based on Carbopol and HPMC did not affect the physicochemical properties of the gels.

Similar results were obtained in a study by Jain et al., where the in situ autoclaving of Carbopol 980NF (0.5%) and HPMC (2%)-based gel had no effect on the pH or viscosity of the drug [80]. The combination of Carbopol with HPMC was stable when sterilized by autoclaving and gamma irradiation. This may have been due to both the effect of EDTA, which prevented oxidative degradation of HPMC, and the low hydroperoxide content of the starting material. The stability of the composition may also have been due to the formation of hydrogen bonds and cross-linking of the polymers. Nevertheless, additional studies on the stability of the combination of Carbopol and HPMC during sterilization by gamma irradiation are needed.

Autoclaving of the in situ gel based on Carbopol 934P (0.35%) and gellan gum (0.55%) did not affect the appearance, transparency or pH of the composition [81]. These results were consistent with the work of Patel et al. [82]. This was expected, since both polymers were stable during autoclaving.

Gel based on gellan gum and MC showed stability after autoclaving and no change in viscosity was observed [83]. In a study [84], in situ autoclaving of HPMC (0.5–0.7%) and gellan gum (0.3–0.5%)-based gels also did not affect the appearance, pH, gelling ability or viscosity of the formulations [84]. The stability of the combination of gellan gum and cellulose derivatives during autoclaving seemed to be due to the low hydroperoxide content of the starting raw materials. In addition, gellan gum, as an anionic polysaccharide, oriented water molecules around itself due to electrostatic forces, resulting in fewer water molecules binding to the HPMC, making it less susceptible to hydrolysis. It also facilitated hydrophobic interactions between the cellulose chains. In addition, hydrogen bonding between the hydroxyl groups of gellan gum and HPMC was possible, providing gel strength and stability during sterilization.

The stability after the in situ autoclaving of a gel based on gellan gum and chitosan was demonstrated in a study [85]. The comparison of UV spectra showed the absence of physicochemical reactions [85]. In this case, it is difficult to say with certainty that this combination of polymers was stable, since the stability of the gel was monitored by only one parameter. However, it can be assumed that the stability was due to electrostatic interactions between the amino groups of chitosan and the carboxyl groups of gellan gum, which ensured the formation of a stable gel [24].

In the study by Balasubramaniam et al., in situ gels of gellan gum and its combination with sodium alginate and sodium citrate (20%) were autoclaved. In formulations containing gellan gum alone, there was no change in viscosity after sterilization, but the addition of alginate or citrate resulted in a decrease in viscosity. The in situ autoclaving of a gel based on gellan gum and alginate resulted in a 10–15% decrease in viscosity [86]. The unstable gel based on gellan gum and alginate could have been attributed to the fact that alginate undergoes hydrolysis during autoclaving.

In a study by Dasanakoppa et al., an in situ gel based on cationic guar gum and hydroxypropyl guar gum showed greater stability than a formulation containing guar gum alone. This combination increased the hydration rate and improved the interaction of the polymers with each other, which improved the physical stability during autoclaving [55].

In the study by Dasankoppa et al., in situ gels based on hydroxypropyl guar gum and its combination with sodium alginate and Carbopol 940 were autoclaved. Formulations containing hydroxypropyl guar gum alone (0.75%), a combination of hydroxypropyl guar gum (0.5%) with Carbopol (0.25%) or hydroxypropyl guar gum (0.25%) with Carbopol (0.125%) and sodium alginate (0.125%), showed a decrease in viscosity after sterilization, but gels containing hydroxypropyl gum (0.5%) with sodium alginate (0.25%) and hydroxypropyl guar gum (0.5%) with sodium alginate (0.125%) and Carbopol (0.125%) showed an increase in viscosity [63]. Accordingly, the combination of hydroxypropyl guar gum with sodium alginate or hydroxypropyl gum (0.5%) with Carbopol (0.125%) and sodium alginate (0.125%) was the most reasonable [87]. These results are at variance with previous studies because Carbopol is stable under sterilization and stabilizes other polymers, whereas alginate undergoes hydrolytic cleavage during autoclaving and requires protection. Further studies are needed to determine the exact conclusions.

In another in situ study, polyacrylic acid and polycarbophil gel were autoclaved at 121 °C for 20 min. Sterilization had a negligible effect on the viscosity and pH of the drug. After autoclaving, precipitation of the polymers was observed, but, after overnight storage under room conditions, the original transparency was restored [88]. This can be explained by the fact that Carbopol, which is similar in structure to polyacrylic acid, and polycarbophil are stable during autoclaving.

### 3.5. Protection of ISS Polymer Complexes from the Effects of Sterilization

#### 3.5.1. Autoclaving

The effect of autoclaving on polymer stability can be reduced by the addition of electrolytes. In the study by Bindal et al. [53], the sterilization of 0.025%, 0.1% and 0.5% solutions of xanthan gum were performed with the addition of sodium chloride, which allowed the viscosity value of the analyzed samples to be maintained.

Xanthan gum was unstable to autoclaving and, at 65 °C, the transition from the double helix conformation to a disordered single chain conformation occurred. However, in the presence of 0.1 M NaCl, the transition temperature increased to 110 °C. This effect was due to the fact that the transition temperature depended on the ionic strength. The stabilizing effect of sodium chloride depended on the salt concentration. No protective effect was observed when 0.01 M NaCl was added to 0.5% xanthan gum solution, but increasing the content to 0.10 M NaCl significantly reduced the loss of apparent viscosity. At the same time, the addition of 0.5 M NaCl caused an increase in viscosity, the reason for which is unknown. The addition of sodium chloride to xanthan gum solutions after autoclaving also resulted in a recovery of apparent viscosity. The degree of recovery depended on the salt concentration and the time of thermal exposure [54].

The effect of electrolytes on the stability of polymers after sterilization by autoclaving was investigated in another study. Hydrogels with the addition of succinate buffer and sodium chloride were compared with samples prepared in purified water. CMC and HPMC showed an increase in viscosity, hardness, compressibility and adhesiveness in the medium with higher ionic strength. The ionic stabilizing effect was also observed with different grades of carbomers [48].

The stabilization of chitosan requires special attention due to its high instability. One possible way to protect chitosan from the effects of thermal sterilization is the addition of polyols. In a study by Jarry et al., triethylene glycol, glycerol, sorbitol, glucose and poly(ethylene glycol) (PEG) were added to chitosan solutions. It was found that all polyols had a protective effect by reducing the viscosity loss caused by autoclaving and had a positive effect on the gelation and gel compression properties. Glucose should not be added to chitosan solutions during autoclaving because it significantly increased solution viscosity and gel strength, but caused color changes due to the reaction of glucose with amino groups (Maillard reaction). PEG was the most effective in protecting chitosan from hydrolysis. The mechanism of stabilizing action is the formation of a protective hydrate layer around the chitosan chains, which prevents hydrolysis and the possible cross-linking of the chains through hydrogen bonds. It is also important to note that polyols had no stabilizing effect on chitosan solutions when sterilized by gamma irradiation. Loss of viscosity was observed regardless of the presence of additives, even when sterilization was performed with frozen solutions [28].

The stabilization of chitosan is possible by the ionic cross-linking of positively charged chains with negatively charged components. Both low molecular weight substances (citrate, sulfate, phosphates) and polymers (pectin, alginate, xanthan gum, carbomer, polycarbophil, poloxamers) can be used for this purpose. The increased stability of chitosan polyelectrolyte complexes is due to the fact that the negatively charged polymer prevents the protonation of chitosan amino groups. In addition, anions buffer the solution and slow down the hydrolysis process [24].

The stability of chitosan is influenced by the degree of deacetylation. In a study by Schuetz et al., autoclaved solutions of type 1 chitosan with a degree of deacetylation of 59% and type 2 chitosan with a degree of deacetylation of 63% were used. The composition prepared with autoclaved type 1 chitosan showed a strong decrease in elastic moduli and viscosity compared with the non-autoclaved composition. The gel based on autoclaved type 2 chitosan retained its properties. Also, the decrease in Mw was more significant in the case of type 1 chitosan. Nevertheless, both formulations showed thermally induced gelation, so autoclaving can be used to sterilize these solutions. However, in cases where it is important to preserve the gelling properties as much as possible, chitosan with a higher degree of deacetylation should be used [89].

To protect HPMC from the action of free radicals generated during autoclaving, EDTA is effective. It binds 2-valent and 3-valent metal ions, which play a key role in the formation of free radicals [40]. Thus, autoclaving 3% HPMC gel with EDTA minimized the loss of viscosity in the presence of 100 ppm H_2_O_2_. The effect was more pronounced at higher peroxide concentrations [41]. Methionine can also be used to protect HPMC, reducing free radical formation during autoclaving [40]. EDTA may be effective in protecting other polymers that may oxidize during autoclaving (poloxamers, CMC), especially in combination with another antioxidant capable of scavenging free radicals (ascorbic acid, methionine) [40,90,91].

#### 3.5.2. Gamma Radiation

As mentioned above, one way to protect the ISS from gamma irradiation may be gamma sterilization in the presence of dry ice. Freezing solutions inhibits the diffusion of free radicals so that they can only react with molecules in close proximity [17,92]. However, this method has less effect on hydrated electrons, which can diffuse and interact with molecules of the substance due to the tunneling effect [92].

Gamma irradiation in the presence of dry ice was used to sterilize in situ terbinafine hydrochloride nanoemulsion gels based on gellan gum. This method had a negligible effect on drug release rates [93].

Another study compared the effect of sterilization on the stability of xyloglucan hydrogels by gamma irradiation methods at room temperature and in dry ice. It was found that gamma irradiation at room temperature caused a loss of gelling properties of the polymer, while sterilization by gamma irradiation in dry ice preserved these properties [38]. Irradiation of CMC solution at −70 °C helped to preserve the viscosity by 66.5% at an irradiation dose of 30 kGy [38].

Irradiation of frozen solutions of metoclopramide and metoprolol protected the drugs from significant degradation observed during sterilization at room temperature (25% loss for metoclopramide and 95% loss for metoprolol) [92].

Another approach may be the addition of radioprotectants that bind free radicals and other active particles, thereby protecting the drug from degradation [17,94]. Such substances may include mannitol, which reacts with the hydroxyl radical, and nicotinamide and pyridoxine, which react with both the hydroxyl radical and the hydrated electron. When metoclopramide and metoprolol were sterilized using these radioprotectants, the recovery was more than 90% at an irradiation dose of 15 kGy, and the levels of radiolytic products were significantly reduced [94].

However, there are very few studies using radioprotectants to protect gel-forming polymers. In one study, ethanol was used to protect a Carbopol-based gel from the effects of gamma irradiation at a dose of 3 Mrd (Megarad = 10 kGy) at 25 °C. After sterilization, the formulations retained pseudoplastic behavior and viscosity loss was reduced to 20%. However, the addition of ethanol to the CMC-based gel was not effective in protecting the gel structure [37].

The use of cetylpyridinium chloride to protect hyaluronic acid gel from gamma irradiation was successful. The protection mechanism is based on the effects of inter- and intramolecular energy transfer leading to a decrease in the number of free radicals in the system [95].

Bazafkan et al. investigated the possibility of protecting solutions of sodium alginate, xanthan gum and CMC from gamma irradiation by adding mannitol and/or ascorbic acid. The addition of mannitol at a concentration of 15% partially reduced the loss of viscosity due to sterilization, which is due to the ability of mannitol to bind OH radicals. However, complete protection was not achieved, which may have been due to the fact that mannitol radicals react with alginate. Similar results were obtained when xanthan gum was sterilized. The addition of ascorbic acid enhanced the protective effect of mannitol due to its ability to bind mannitol radicals. When CMC was sterilized with mannitol, the viscosity of the sample was less than 500 cps, while for the CMC/mannitol/ascorbic acid solution, the decrease in viscosity was not as significant (from 180,000 cps to 8000 cps). This confirms the protective effect of ascorbic acid and the degradation of the polymer under the action of mannitol radicals [96].

When the CMC solution was irradiated with vitamin C, the relative viscosity of the solution was 38% compared with the unirradiated solution. Ascorbic acid changed the pH, which decreased the reaction rate of OH radicals and CMC [38].

In another study, vitamin C and histidine were used as radioprotectors in the sterilization of xyloglucan hydrogel, but they did not provide protection against irradiation [17].

### 3.6. Methods for Screening the Stability of the ISS Following Sterilization

The majority of authors have recommended that the stability of the ISS be investigated following sterilization by a number of parameters, including the appearance and transparency of the material, its pH, viscosity, gelation ability and the characteristics of the gelation process.

#### 3.6.1. General Appearance and Transparency

The general appearance and transparency of the sol and gel are determined through visual inspection on both white and black backgrounds. It is imperative that the composition retains its transparency following the sterilization process. A change in color may be indicative of the degradation of the polymer or other in situ gel components [28,32,56]. In the context of the ISS, it is possible for polymer deposition to occur following the application of autoclaving. However, the composition in question will regain its transparency following a period of cooling. Consequently, it is recommended that the determination of transparency be conducted after a suitable interval at room temperature [78,88].

#### 3.6.2. Gel pH

The pH of the gel is quantified using a digital pH meter. Additionally, the potentiometric method can be employed to ascertain the indicator, which is based on measuring the potential difference between electrodes immersed in the solution under study and a reference solution [26]. It is possible that a decrease in pH may occur in the case of Carbopol following the sterilization process. However, this indicator is restored to its initial value after a period of three weeks [14].

#### 3.6.3. Viscosity

Viscosity is a crucial parameter that provides insight into the alterations in the polymer structure and properties following sterilization. Polymer degradation, which occurs during autoclaving or under the influence of gamma irradiation, is typically accompanied by a reduction in gel viscosity [18,25,28,51]. In the majority of studies, the viscosity (mPa·s) of the sol and gel is recorded using a Brookfield viscometer as a function of the shear rate (s^−1^) applied to the composition.

#### 3.6.4. Gelation Ability

The gel-forming ability of an ISS is the primary determinant of its efficacy as a drug delivery system. The suitability of a given sterilization method is contingent upon the ability of the system to form a gel. The gel-forming ability of a formulation is determined by placing it in a medium with a specific pH and ionic composition (for ophthalmic formulations, simulated tear fluid (STF) with a pH of 7.4 is used) at the gelation temperature. The strength of the gel that is formed and the gelation time are then visually evaluated [32,74].

#### 3.6.5. In Situ Gelation Characteristics

Changes in gelation temperature subsequent to sterilization have the potential to result in the loss of a critical property of an ISS, namely its capacity to undergo a phase transition at the injection site. The gelation temperature can be determined through the implementation of a variety of methodologies. In the inverted tube method, the tube containing the sample is placed in a continuously heated water bath and tilted at regular intervals to measure the phase transition temperature. The gelation temperature is defined as the temperature at which the sample ceases to flow. Another method is to gradually heat the sample while continuously stirring with a magnetic stirrer, noting the temperature at which the magnetic rod ceases movement as a result of the gelation process [26,29]. These methods are distinguished by their rapid analysis time but relatively low accuracy [29].

The gelation temperature can be determined using a rotational viscometer, whereby the sample is continuously heated and the temperature at which a significant increase in viscosity is observed is recorded [26,29]. Nevertheless, if the temperature is increased too rapidly, the phase transition temperature may be incorrectly determined [29]. A more accurate method for measuring the elastic modulus (G′) and loss modulus (G″) is through the use of a rheometer. The measurements are conducted in the mode of small oscillations with a frequency of 1 Hz, while the composition is subjected to continuous heating. The resulting values are then plotted on a rheological curve. The gelation temperature can be identified as the point of intersection between G′ and G″. In addition to the gelation temperature, rheological curves facilitate the characterization of the viscoelastic properties of the gel [28,29,89].

In the case of pH-sensitive polymers, the pH at which gelation occurs is identified. The composition is placed in a beaker, and sodium hydroxide is added dropwise. The pH is monitored with a pH meter, and the viscosity is determined. The pH value at which a pronounced increase in viscosity is observed is defined as the gelation pH [6].

The gelation time is determined by inverting the tube at intervals with the sample heated to the gelation temperature. The time required for the transition from sol to gel is recorded [5].

In a number of studies, some additional characteristics have been determined for experimental samples subjected to sterilization.

The processes of the degradation or cross-linking of polymer chains can be assessed by the value of Mw. This index is determined for chitosan-based ISSs by gel permeation chromatography [25]. An alternative method is flow fractionation in an asymmetric flow field combined with multi-angle light scattering [89].

For chitosan gels, mechanical properties are evaluated using the MACH-1™ micromechanical tester. Sample thickness is measured using a high-precision actuator, and stress-relaxation values at maximum and equilibrium load and relaxation time are determined by compression with a cylindrical indenter [25,28].

The sterilization of the samples may also affect the release of the drug from the delivery system. The release profile is evaluated using Franz diffusion cells or the dialysis bag method. A Franz diffusion cell contains a polymeric membrane that separates the donor compartment containing the sample to be analyzed from the receptor compartment containing the medium. The drug diffuses through the membrane into the receptor compartment, from which aliquots are taken at intervals and the percentage of release is determined [26].

For most ISSs, the drug content after sterilization is determined by various methods, mainly spectrophotometric methods. A high percentage of drug content after sterilization indicates stability and an absence of interaction between the in situ gel components. For the same purpose, UV-visible and IR spectra can be recorded, and the absorption spectra of sterilized and unsterilized formulations can be compared with the absorption spectra of the pure drug and the placebo, and the position and color of the spots can be determined.

Thus, based on the above research methods and techniques, a protocol for the study of ISS stability after sterilization with a minimum of five items can be formed.

## 4. Discussion

Autoclaving and gamma irradiation are the most commonly industrially used sterilization methods for excipients and pharmaceutical compositions, along with chemical sterilization with ethylene oxide, shown for instruments and packaging materials, and are the only ones approved by regulatory authorities.

Among the most commonly used methods of sterilization of polymer compositions, there are currently no fully universal solutions that do not reliably affect the properties of polymers. For ISSs, highly sensitive to any changes in the qualitative and quantitative composition of compositions, capable not only of changing physicochemical properties after sterilization but also of complete loss of functionality (stimulus sensitivity, ability to form a gel in situ), a comprehensive study of the effects of sterilization, assessment of the risks of such results and the ability to effectively manage these risks is essential.

The analysis of the published results showed that the risk of changes in physicochemical properties after sterilization for in situ compositions depends to a greater extent not on their leading gelling stimulus (thermo/ion/pH sensitivity) but on the structural features of a particular polymer. For example, many studies have reported instability upon the autoclaving of chitosan derivatives (thermosensitive and pH-sensitive) and various gums (ion-sensitive). Natural polymers are more often sensitive to various sterilization methods.

Thus, the decision regarding the need to protect the polymer composition from the effects of sterilization and the choice of an effective and safe method of sterilization should be based on factors such as the nature of the polymer, the structure of the polymer, the viscosity of the solution (composition before in situ transition) and the safety of the chosen method of sterilization for the active substance and other components of the drug.

In general form, the adoption of the decision can be illustrated in a flowchart (Figure 5).

After a compilation of published experimental results and their analysis, it can be concluded that the solution to the risks associated with ISS sterilization lies in the area of reasonable selection of the most stable stimuli-sensitive polymer from the pool/use of a protector, and the definition of critical parameters regarding the protocol before and after sterilization.

It is possible to identify the main protectors that positively influence the stability of compositions after sterilization (Table 2). The mechanisms of their protective action are diverse.

The mechanism of protection of chitosan chains by polyols proposed by Jarry et al. is similar to the way osmolytes stabilize proteins under thermal stress [97,98]. Due to the preferential exclusion effect, a thin hydration shell is formed on the protein surface around which polyol molecules are grouped. This effect is due to the fact that the radius of polyol molecules is larger than that of water molecules. Polyols organize the structure of water molecules by increasing the water–water hydrogen bonds. This leads to the dehydration of the protein surface, making biomolecules less conformationally flexible and more stable. Hydrogen bonds between the osmolyte and the protein also contribute to stabilization [97,98]. The protective effect depends on the molecular volume of the polyol, i.e., the larger the molecular size, the greater the degree of preferential hydration [97]. This is in agreement with the results of Jarry et al., where PEG was found to be the most effective in protecting chitosan.

The presented mechanism of action also underlies the gelation of chitosan with glycerophosphate, which forms a protective shell and prevents the interaction of chitosan chains. When the phase transition temperature is reached, a proton transfer from the chitosan to the gelator occurs, the cohesion of the polyol protective shell decreases and the polymer chains interact to form a three-dimensional network structure [99].

It can be assumed that, during the sterilization of chitosan, polyols order the structure of water molecules in the hydration shell, and the formation of hydrogen bonds between water molecules and between polyol and chitosan provides stability to the polymer chains (Figure 6). Not only the polyols used in the study of Jarry et al. but also other osmolytes can have this effect, e.g., polyols (xylitol, trehalose, sucrose); methylamines (trimethylamine-oxide, betaine); amino acids (taurine) [97,100].

The study by Adams et al. did not provide an explanation for the mechanism of Carbopol stabilization by ethanol, but this effect seems to be due to the ability of ethanol to bind hydroxyl radicals [101,102]. Another solvent, dimethyl sulfoxide, is also an effective scavenger of -OH radicals, but its high toxicity limits its use [102,103].

Mannitol is also capable of reacting with the hydroxyl radical, but the interaction products may be reactive and may interact with the polymer [96].

Ascorbic acid may be a promising compound for protecting ISSs from gamma irradiation due to its high activity in scavenging reactive oxygen species to form the less reactive ascorbate radical and dehydroascorbic acid [104]. Ascorbic acid competes with oxygen for intermediate target radicals and is able to bind the mannitol radical, thereby enhancing the protective effect of mannitol [96,105].

Histidine, as a scavenger of hydroxyl radical (OH) and singlet oxygen, may also be effective in defense against gamma irradiation. Histidine is able to absorb a large amount of gamma ray energy with the formation of non-toxic products, mainly aspartate [105]. Carnosine, which is composed of beta-alanine and histidine residues, has similar properties [106].

Although ascorbic acid and histidine had no protective effect on xyloglucan hydrogel, they may be effective in stabilizing other polymers [17].

Nicotinamide and pyridoxine, which protected metoclopramide and metoprolol from gamma irradiation due to their ability to react with the hydroxyl radical and hydrated electron, are also potential radioprotectors for ISSs [94].

No studies were found using thiol-containing compounds to protect gel-forming polymers from gamma irradiation, but they may prove to be effective radioprotectors for the ISS. SH compounds (glutathione, cysteine, N-acetylcysteine) readily react with free radicals and prevent oxidative degradation by scavenging reactive oxygen species [107,108,109,110,111,112]. Thiol-containing compounds are being actively studied as tissue and cell radioprotectors by preventing radiation-induced DNA breaks [108]. Among thiol-containing compounds, N-acetylcysteine is the most convenient to use due to its availability and water solubility and its antibacterial properties [103,110]. In the sterilization of bone allografts, N-acetylcysteine helped to reduce radiation-induced damage by reacting with hydroxyl radicals and H_2_O_2_. However, this effect is dependent on radiation dose and N-acetylcysteine concentration, so it is important to select optimal conditions [103].

## 5. Conclusions

Sterilization is a necessary technological step for obtaining ISS-based drugs intended for ophthalmic, parenteral or wound application. At the same time, the issue of stability of polymeric stimuli-sensitive compositions after sterilization has been insufficiently studied for a long time, which certainly complicates the transition from the in situ composition development stage to the development of formulations for clinical trials or technology transfer.

The solution to the problem of low stability of polymeric in situ compositions lies in the identification, evaluation and management of risks arising from the sterilization process. To create specific recommendations, it was necessary to conduct a comprehensive review, research and analysis of published experimental data answering the following concerns: the influence of different sterilization methods on the stability of stimuli-sensitive and physicochemical characteristics of polymer compositions; the mechanisms of instability cases; the different relation of stimuli-sensitive polymers to sterilization and the justified choice of stable excipients; managing the stability of compositions by adding protective agents and a list of tests required for quality control before and after sterilization.

Autoclaving and gamma irradiation can be considered as alternative sterilization methods suitable for sensitive systems. To protect polymers from the effects of sterilization, the introduction of protectors (polyols, antioxidants, alcohols) or the use of additional technological approaches (cryogenic treatment) can be recommended.

At the same time, among polymers sensitive to stimuli, there are the most stable excipients (poloxamers, gellan gum, carbomers), in favor of which it can be recommended to choose when developing ISSs of certain stimuli.

To evaluate the quality and prove the absence of sterilization effects on ISS properties, the implementation of a universal protocol consisting of at least five points—appearance and transparency, pH, viscosity, gelling ability and in situ gelling characteristics—can be recommended.

Thus, the recommendations and approaches presented in this review, based on published experimental data, can effectively manage the risks associated with the development of sterile ISSs, appearance and transparency, pH, viscosity, gelation capacity and in situ gelation characteristics. These practical recommendations are likely to contribute to the development of more sterile preparations and medical devices based on in situ technology in the future, when the problem of loss of quality and properties of formulations after sterilization is finally overcome.

## Figures and Tables

**Figure 1 polymers-16-02943-f001:**
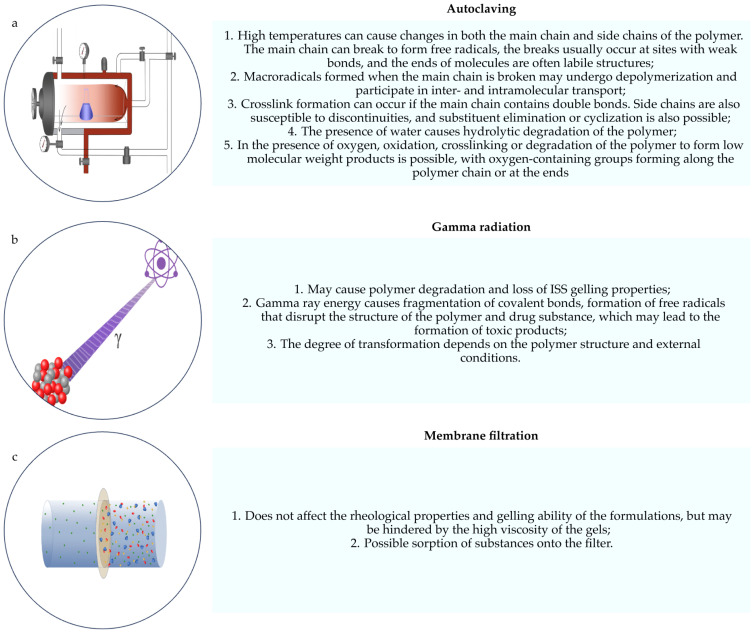
Risks occurring when using the sterilization method: (**a**) autoclaving scheme; (**b**) radiation sterilization scheme; (**c**) membrane filtration.

**Figure 2 polymers-16-02943-f002:**
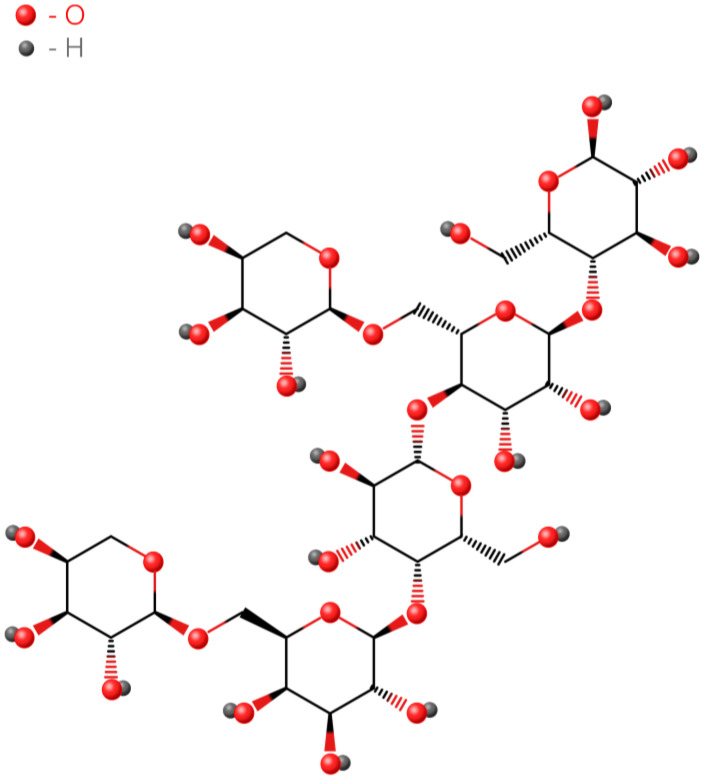
Xyloglucan formula.

**Figure 3 polymers-16-02943-f003:**
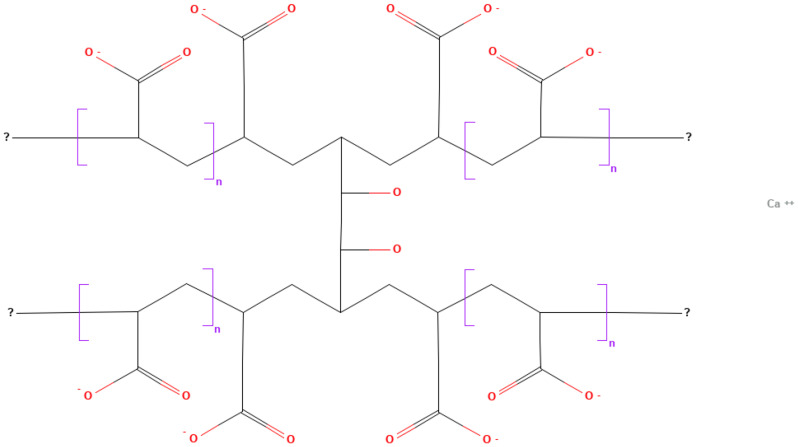
Polycarbophil formula (obtained from PubChem).

**Figure 4 polymers-16-02943-f004:**
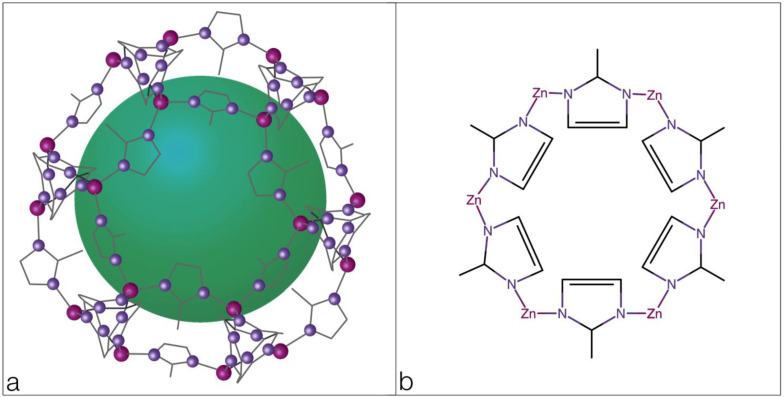
Zeolites: (**a**) 3D formula; (**b**) structural formula.

**Figure 5 polymers-16-02943-f005:**
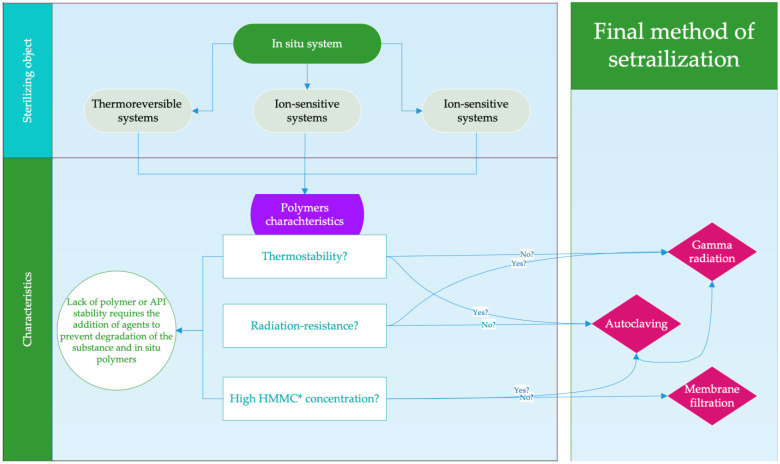
Decision-making flowchart for selecting a sterilization method. *—high molecular mass compound.

**Figure 6 polymers-16-02943-f006:**
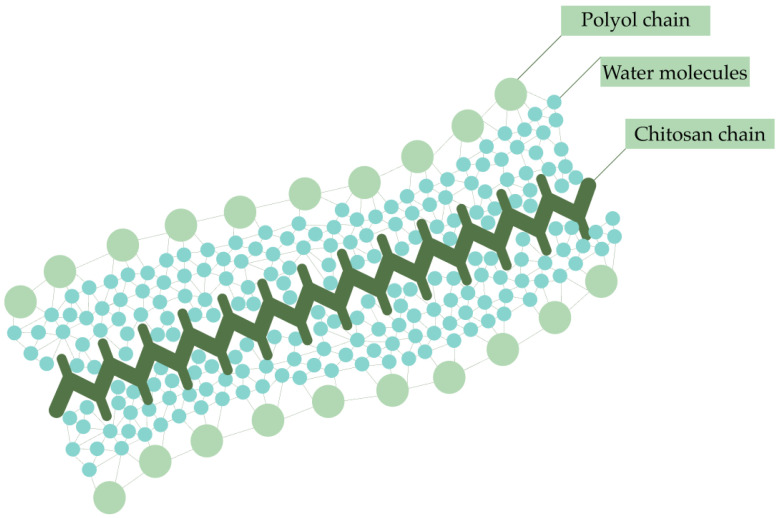
Chitosan protection mechanism using polyol molecules.

**Table 1 polymers-16-02943-t001:** Characterization of the main ISS sterilization methods.

Sterilization Method	Advantages
Autoclaving	The method is well researched and widespread; High efficiency; Convenient for large-scale drug production; Can be used for a wide range of thermostable drugs.
Gamma radiation	High efficiency of the method; No heating during sterilization; This methodology permits uninterrupted operation while providing precise quantification of the absorbed radiation dose.
Membrane filtration	Does not require exposure to external physical and chemical factors for sterilization; Safe method; Does not affect polymer compositions and APIs.

**Table 2 polymers-16-02943-t002:** Effective combination of stimuli-sensitive polymers and protective agents for sterilization stability.

Polymer	Sterilization Method	Protective Agent	Reference
Xanthan gum	Autoclaving	Sodium chloride	[54]
CMC HPMC Carbopol	Autoclaving	Succinate buffer and sodium chloride	[48]
Chitosan	Autoclaving	Triethylene glycol, glycerin, sorbitol, glucose, PEG	[28]
HPMC	Autoclaving	EDTA	[41]
Carbopol	Gamma radiation	Ethanol	[37]
Hyaluronic acid	Gamma radiation	Cetylpyridinium chloride	[95]
Sodium alginate Xanthan gum	Gamma radiation	Mannitol	[96]
Xanthan gum CMC	Gamma radiation	Mannitol and ascorbic acid	[96]
CMC	Gamma radiation	Ascorbic acid	[38]

## Data Availability

The data presented in this study are openly available in the article.

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
