# Peer review of "Role of Sterilization on In Situ Gel-Forming Polymer Stability"

_polymers, 2024, doi:10.3390/polym16202943_

Round 1

Reviewer 1 Report

Comments and Suggestions for Authors

The manuscript «Role of sterilization on gel forming polymers stability» submitted by Bakhrushina E.O. et al. to Polymers is devoted to the review of the sterilization methods of gel forming polymers and their influence to the polymers stability. In introduction authors described the gel-forming in situ systems (ISS), approaches for their sterilization and problems with their usage. Review contains two main section Modern Sterilization Methods and Risks for the ISS and Basic Polymers for ISS development. Section Discussion is devoted to the comparison of sterilization methods and approaches for protection of polymers and active pharmaceutical ingredients (APT) on sterilization conditions. Conclusions are corresponded to review content. Manuscript contains 2 Tables and one figure. 

The manuscript contains some of drawbacks and weakness and is need to be revised.

I have questions and remarks:

1)      Please check the references (13, 37,86, 14, 23, 24, 26, 50, 51). Doi are absent or formatting is not correspond to the requirements.

2)      Abbreviation ISS used for in situ systems and international space station. Please use this abbreviation for one of these terms.

3)      Line 614. For irradiation description authors used Gy and Mrd. Please decode this abbreviation (Mrd) and give the comparison in Gy.

4)      Please check the phrase on 146-147 lines. PEO are used three times, is it correct? I don’t understand the idea.

5)      Please check the phrase on 151 line. I think authors missed something.

6)      Lines 172-176. What drug was used in work [32]. Please add the name in the text.

7)      Line 361. What does you mean by autoclave stability. Please check the English.

8)      Line 839. There is a point in the central part of sentence. Please check it.

9)      In the reference (Smth et al.) please use the point in all cases in the text.

10)  Table 1. Please add the additional line to restrict the table cells if it is possible. It was complicated for me to find the parameters for autoclaving or gamma irradiation.

The mistypings are present in the text.

Comments on the Quality of English Language

The mistypings are present in the text. Moderate editing of English language required.

Author Response

Comment 1: Please check the references (13, 37,86, 14, 23, 24, 26, 50, 51). Doi are absent or formatting is not correspond to the requirements.

Response 1: Thank you for your comment! We’ve revised references.

Comment 2: Abbreviation ISS used for in situ systems and international space station. Please use this abbreviation for one of these terms.

Response 2: Sorry for misprint. We’ve changed the abbreviation.

Comment 3: Line 614. For irradiation description authors used Gy and Mrd. Please decode this abbreviation (Mrd) and give the comparison in Gy. 

Response 3: Thank you! The explanation is added now.

Comment 4: Please check the phrase on 146-147 lines. PEO are used three times, is it correct? I don’t understand the idea. 

Response 4: Thank you! The phrase is revised.

Comment 5: Please check the phrase on 151 line. I think authors missed something.

Response 5: Yes, you are right, the phrase was corrected. Thank you!

Comment 6: Lines 172-176. What drug was used in work [32]. Please add the name in the text. 

Response 6: The drug INN is added. Thank you for your comment!

Comment 7: Line 361. What does you mean by autoclave stability. Please check the English.

Response 7: The sentence is corrected. Thank you!

Comment 8: Line 839. There is a point in the central part of sentence. Please check it.

Response 8: The point is changed to comma. Thank you! 

Comment 9: In the reference (Smth et al.) please use the point in all cases in the text. 

Response 9: All references are corrected in text. Thank you!

Comment 10: Table 1. Please add the additional line to restrict the table cells if it is possible. It was complicated for me to find the parameters for autoclaving or gamma irradiation.

Response 10: We’ve changed the table and added another figure. Now it’s clear what are the advantages and risks for most commonly used sterilization methods. Thank you!

Reviewer 2 Report

Comments and Suggestions for Authors

Elena O. Bakhrushina et al. have reviewed the issues of risk assessment and resolution in the sterilization of stimuli-sensitive systems, as well as ways to stabilize such compositions, whcih provides recommendations and approaches, formulated on the basis of published experimental data, that allow effective management of risks arising during the development of in situ systems requiring sterility. The manuscript however requires further modification to improve its presentation for a wider readership of the journal. The specific comments are as follows:

[1] In Table 1, the characterization of the main ISS sterilization methods should also include the features and advantages.

[2]  In the section Basic polymers for ISS development, the section 3.4. In situ polymer compositions seems to be improper.

[3] The authors are suggested to provide some suggestions on how to choose sterilization nethods according to the  stability needs of the ISS following sterilization.

[4] The future prospect is necessary.

[5] Some literatures are suggested to add (Exploration 2023, 3, 20220173. https://doi.org/10.1002/EXP.20220173; Exploration 2023, 20210117. https://doi.org/10.1002/EXP.20210117;).

Author Response

Comment 1: In Table 1, the characterization of the main ISS sterilization methods should also include the features and advantages.

Response 1: Thank you for the comment! We’ve changed the first table and added new figure. Now it’s clear what are the advantages and risk for chosen methods. 

Comment 2: In the section “Basic polymers for ISS development”, the section “3.4. In situ polymer compositions” seems to be improper.

Response 2: We’ve changed the name of section to make in more logical.

Comment 3: The authors are suggested to provide some suggestions on how to choose sterilization nethods according to the  stability needs of the ISS following sterilization.

Response 3: We’ve added new paragraphs in discussion section and developed the flowchart for selecting a sterilization method. Thank you for the comment!

Comment 4: The future prospect is necessary.

Response 4: We’ve made some conclusions and described ideas in discussion. Thank you!

Comment 5: Some literatures are suggested to add (Exploration 2023, 3, 20220173. https://doi.org/10.1002/EXP.20220173; Exploration 2023, 20210117. https://doi.org/10.1002/EXP.20210117;).

Response 5: We’ve added these references. Thank you for the advice!

Reviewer 3 Report

Comments and Suggestions for Authors

Author Response

Comment 1: ISS in line 26 is not a common abbreviation.

Response 1: Thank you for the comment! We’ve corrected this mistake.

Comment 2: Line 37, all in situ forming gels are not mucoadhesive. It depends on the composition of the gels.

Response 2: Thanks for the comment! Changed the wording and made it more correct.

Comment 3: Protecting agents mentioned in table 2 are not practically applicable. For example, sodium chloride cannot be used in chitosan as wound dressing. Ascorbic acid is not stable compound and is oxidized easily.

Response 3: As you can see in this table we didn’t even mentioned sodium chloride as a protective agent for chitosan. This excipient can be used for other dosage forms and in situ polymers. Regarding ascorbic acid - the mechanism by which ascorbic acid exerts a protective effect is through its ability to bind to free radicals, thereby oxidizing itself. This process serves to safeguard the drug from gamma-irradiation.

Comment 4: In figure 1 polyol chains look much bigger compared to chitosan chain. Chitosan repeat unit is much bigger than polyol. This images is not in accordance with the reality.

Response 4: Thank you for the comment! The figure is corrected.

Comment 5: Toxicity of degradation products released from polymers after application of different degradation methods are not discussed.

Response 5: Thanks for the comment! The reasoning on this topic does not make much sense, since polymers lose their gelation properties after degradation and are no longer of any scientific relevance for analyzing as in situ systems. Studying the toxicity of degradation products after degradation was not the original purpose of this review and makes more sense in a research article. 

Reviewer 4 Report

Comments and Suggestions for Authors

- please provide the chemical structure for each polymer, there is no suggested mechanism or general mechanism that will give high potential to your work and will help many readers

please provide the suggested chemical mechanism for each method to explain the difference and will be helpful if you conclude these methods provide the best method with each gel

- some items have enough data and some have concise data, such as ph thermosensitive polymer please revise and complete missed that to have an excellent collective organized data

- you have to make a future outlook

Author Response

Comment 1: please provide the chemical structure for each polymer, there is no suggested mechanism or general mechanism that will give high potential to your work and will help many readers

Response 1: Thank you for the comment! We’ve added new figures.

Comment 2: please provide the suggested chemical mechanism for each method to explain the difference and will be helpful if you conclude these methods provide the best method with each gel

Response 2: The discussion section has been expanded and chemical formulas have been added to understand the mechanism of the effects of sterilization methods on polymers.

Comment 3: some items have enough data and some have concise data, such as ph thermosensitive polymer please revise and complete missed that to have an excellent collective organized data

Response 3: We have expanded the mentioned section, added a large number of figures and redesigned the tables. We have also added new conclusions to the discussion section. Thank you very much for your comments!

Comment 4: you have to make a future outlook

Response 4: We’ve added new ideas and conclusions in 4th and 5th sections. Thank you for the comment!

Reviewer 5 Report

Comments and Suggestions for Authors

General comments

This article is intended as a review of effect of sterilization on the stability of gel-forming systems based on different stimuli and polymers and to identify effective approaches for stabilization of compositions requiring sterilization. Sterilization of medicaments based on hydrogel is necessary to prevent microbial contamination of the drug product, which may result in degradation of the active pharmaceutical ingredient during storage, as well as the development of infection in the patient's body. Thus sterilization is an indispensable technological step for obtaining drugs intended for ophthalmic, parenteral or wound application.

The relevance of this work is beyond doubt. Identification, assessment and management of risks arising during sterilization will facilitate the transition from the stage of in situ composition development to the development of formulations for clinical trials or technology transfer. Accordingly, a good review article analyzing published experimental data on a given problem may be in high demand.

However, the material extended in this work requires a more effective presentation. Therefore, the article is not yet ready for publication. In particular, the following points should be noted:

1) The article contains little illustrative material. Only 2 tables and 1 figure are presented. It is necessary to add color block diagrams, for example, for the classification of sterilization objects, for the classification of different sterilization methods, etc.

2) Since the main sterilization methods for hydrogels are Autoclaving and Gamma radiation, 2 figures can be added, with Schematic representation of an autoclave- and gamma radiation equipment or process diagram used for sterilization.

The closest review paper [11]:

Galante, R.; Pinto, T.J.A.; Colaço, R.; Serro, A.P. Sterilization of Hydrogels for Biomedical Applications: A Review. J. Biomed. Mater. Res. B. Appl. Biomater. 2018, 106, 2472–2492, doi:10.1002/jbm.b.34048

provides 4 figures and 6 tables, so it looks more advantageous.

3) It would also be desirable to insert a table listing the side effects of sterilization reported by various authors, with references to the relevant articles.

Specific comments

4) Line 839. The point after ISS must be replaced with a comma.

5) In many references (1, 8, 15, 22-24, etc.) the year of article publication is not highlighted in bold.

Author Response

Comment 1: The article contains little illustrative material. Only 2 tables and 1 figure are presented. It is necessary to add color block diagrams, for example, for the classification of sterilization objects, for the classification of different sterilization methods, etc.

Response 1: Thank you for the comment! We have expanded the illustrative material. We made it more understandable and informative.

Comment 2: Since the main sterilization methods for hydrogels are Autoclaving and Gamma radiation, 2 figures can be added, with Schematic representation of an autoclave- and gamma radiation equipment or process diagram used for sterilization.

The closest review paper [11]:

Galante, R.; Pinto, T.J.A.; Colaço, R.; Serro, A.P. Sterilization of Hydrogels for Biomedical Applications: A Review. J. Biomed. Mater. Res. B. Appl. Biomater. 2018, 106, 2472–2492, doi:10.1002/jbm.b.34048

provides 4 figures and 6 tables, so it looks more advantageous.

Response 2: In the first picture we have schematically depicted the most commonly used methods of in situ sterilization of polymers and also described the risks of their application in the ISS production process. There are now 2 tables and 6 figures in the article. Thank you very much for your comments!

Comment 3: It would also be desirable to insert a table listing the side effects of sterilization reported by various authors, with references to the relevant articles.

Response 3: All the risks of the description on the figure 1, and the references listed in the description to him. Thank you very much for your comment!

Comment 4: Line 839. The point after ISS must be replaced with a comma.

Response 4: Thanks for the comments! Revisions have been made

Comment 5: In many references (1, 8, 15, 22-24, etc.) the year of article publication is not highlighted in bold.

Response 5: Thanks for the comments! Revisions have been made

Round 2

Reviewer 3 Report

Comments and Suggestions for Authors

as attached

Comments on the Quality of English Language

abbreviations were not corrected such as ISS

Author Response

Response.

Notwithstanding the unfavorable nature of your commentary, our team of authors extends to you its sincere respect and gratitude for your investment of time in the examination of our manuscript.

In light of the aforementioned comments, we urge you to exercise caution and objectivity in your evaluation of the article. As may be observed in the explicitly stated objective of the review, the principal outcome of the paper is the aggregation of data regarding the impact of sterilization techniques on the stability of in situ systems and potential strategies for the protection and stabilization of systems that require sterilization. It was not our intention to develop a novel sterilization method or to synthesize new polymers with resistance to all existing sterilization methods. Furthermore, we would like to add that your comment has prompted us to re-examine the studies cited in our review and evaluate their scientific rigor, comparing them with our own work. Please refer to the table below for a list of articles on in situ sterilization of systems and their limitations.

Reference

Year

Disadvantages

Keywords

Sintzel, M.B.; Merkli, A.; Tabatabay, C.; Gurny, R. Influence of Irradiation Sterilization on Polymers Used as Drug Carriers 1115

- A Review. Drug Dev. Ind. Pharm. 1997, 23, 857–878

1997

- Most attention is paid to the effect of sterilization on polymers in the solid state (powders, microspheres, etc.), few studies are given where gels are the object of study.

- The effect of autoclaving on the stability of gels is not considered.

- No information on the effect of sterilization on the stability of gels of chitosan, hydroxypropylmethylcellulose, xyloglucan, gum, alginate, polycarbophilus.

- The influence of polymers on each other during their co-sterilization is not considered.

- Information on ways to protect formulations from the negative effects of sterilization is not adequately provided (no protective agents that can be used during autoclaving are given, no new potential protective agents are proposed).

- Methods for in situ gel stability screening are not presented.

Hydrogel; sterilization

Galante, R.; Pinto, T.J.A.; Colaço, R.; Serro, A.P. Sterilization of Hydrogels for Biomedical Applications: A Review. J. Biomed. Mater. Res. B. Appl. Biomater.2018, 106, 2472–2492, doi:10.1002/jbm.b.34048

2018

- There is no complete information on the effect of sterilization on the stability of poloxamer, gum, and carbomer gels.

- The influence of polymers on each other during their joint sterilization is not considered.

- Methods to protect formulations from the negative effect of sterilization are not presented (protective agents, mechanism of their protective effect, etc. are not presented).

- Methods of in situ gel stability screening are not presented 1.

Hydrogel; sterilization

S. A. Bento, C.; Gaspar, M.C.; Coimbra, P.; de Sousa, H.C.; E. M. Braga, M. A Review of Conventional and Emerging Technologies for Hydrogels Sterilization. Int. J. Pharm. 2023,634, doi:10.1016/j.ijpharm.2023.122671.

2023

- There is no information on the effect of autoclaving on the stability of xyloglucan, guar gum, carbopol gels and gamma irradiation on the stability of poloxamers, gums, carbomers, xyloglucan, cellulose derivatives.

- The influence of polymers on each other during their joint sterilization is not considered.

- Insufficient information on the methods of protection of formulations from the negative effects of sterilization (incomplete list of protective agents, no detailed description of the mechanisms of their protective effect, etc.).

- Methods for in situ gel stability testing are not presented.

- Insufficient information on the effect of sterilization on pH, gelation ability and gelation temperature.

Hydrogel;

polymers;sterilization

[1] Since the review is devoted to gels for biomedical applications, the most important stability parameters for them are mechanical and rheological properties, so the main focus is on these. For in situ systems it is important to control pH, gelation capacity and gelation temperature, information on the change of these parameters of gels during sterilization is not given in the review.

I am also a bit surprised by your comment "Many polymers can form gels in response to temperature, pH, or other stimuli if they are chemically modified". We all know that any chemical and physicochemical modification can lead to the desired result, but at the same time we cannot describe in the article all chemical compounds potentially capable of stimulus-sensitive phase transitions. This approach has no scientific relevance and often misleads many readers. In our article, polymer groups are logically structured and detailed descriptions are given for them with possible stability changes after sterilization.

We found the comment on the title of the second chapter of our article to be reasonable. Thank you very much! You are right, in this case the sterilization methods should be called "current" instead of "modern".

In addition to the first table, we have listed below the most relevant studies that contain only bits and pieces of knowledge about sterilization of stimulus-sensitive systems. Our review has just collected all such data, allowing a sober assessment of the impact of sterilization on in situ systems and a concrete evaluation of all sterilization methods.

Reference

Year

Data

Keywords

Cooper, R.C.; Yang, H. Hydrogel-Based Ocular Drug Delivery Systems: Emerging Fabrication Strategies, Applications, and Bench-to-Bedside Manufacturing Considerations. J. Control. Release 2019, 306, 29–39.

2019

Characteristics of sterilization methods

Hydrogel; sterilization

Ruel-Gariépy, E.; Leroux, J.C. In Situ-Forming Hydrogels - Review of Temperature-Sensitive Systems. Eur. J. Pharm. Biopharm. 2004, 58, 409–426, doi:10.1016/j.ejpb.2004.03.019.

2004

Chitosan sterilization

In situ gel; sterilization

Kurniawansyah, I.S.; Rusdiana, T.; Sopyan, I.; Farisa, I.; Arya, D.; Wahab, H.A.; Nurzanah, D. Comparative Study of in Situ Gel Formulation Based on the Physico-Chemical Aspect: Systematic Review. mdpi.com 2023, doi:10.3390/gels9080645.

2023

Mention of pH-sensitive gels sterilization

In situ gel; sterilization

Kurniawansyah, I.S.; Rusdiana, T.; Sopyan, I.; Subarnas, A. A Review on Poloxamer and Hydroxy Propyl Methyl Cellulose Combination as Thermoresponsive Polymers in Novel Ophthalmic in Situ Gel Formulation and Their Characterization. Int. J. Appl. Pharm. 2021, 13, 27–31, doi:10.22159/ijap.2021v13i1.39697.

2021

Mention of in situ gels sterilization

In situ gel; sterilization

Kondepati, H.V.; Kulyadi, G.P.; Tippavajhala, V.K. A Review on In Situ Gel Forming Ophthalmic Drug Delivery Systems. Res. J. Pharm. Technol. 2018, 11, 380, doi:10.5958/0974-360X.2018.00069.0.

2018

Mentioning the need to sterilize ophthalmic in situ systems

In situ gel; sterilization

Ahmed, B.; Jaiswal, S.; Naryal, S.; Shah, R.M.; Alany, R.G.; Kaur, I.P. In Situ Gelling Systems for Ocular Drug Delivery. J. Control. Release2024, 371, 67–84, doi:10.1016/j.jconrel.2024.05.031.

2024

Mention of effective in situ sterilization of gellan gum gels

In situ gel; sterilization

Das, B.; Chattopadhyay, D.; Rana, D. The Gamut of Perspectives, Challenges, and Recent Trends for in Situ Hydrogels: A Smart Ophthalmic Drug Delivery Vehicle. Biomater. Sci. 2020, 8, 4665–4691, doi:10.1039/D0BM00532K.

2020

Reference to the problem of stability of in situ gels during sterilization

In situ gel; sterilization

Dewan, M.; Adhikari, A.; Jana, R.; Chattopadhyay, D. Development, Evaluation and Recent Progress of Ocular in Situ Gelling Drug Delivery Vehicle Based on Poloxamer 407. J. Drug Deliv. Sci. Technol. 2023, 88, 104885, doi:10.1016/j.jddst.2023.104885.

2023

Reference to the stability of poloxamers during autoclaving

In situ gel; sterilization

Moon, H.J.; Ko, D.Y.; Park, M.H.; Joo, M.K.; Jeong, B. Temperature-Responsive Compounds as in Situ Gelling Biomedical Materials. Chem. Soc. Rev. 2012, 41, 4860, doi:10.1039/c2cs35078e.

2012

Mention of the possibility to sterilize in situ gels by membrane filtration method

In situ gel; sterilization

ZieliÅ„ska, A.; Soles, B.B.; Lopes, A.R.; Vaz, B.F.; Rodrigues, C.M.; Alves, T.F.R.; Klensporf-Pawlik, D.; Durazzo, A.; Lucarini, M.; Severino, P.; et al. Nanopharmaceuticals for Eye Administration: Sterilization, Depyrogenation and Clinical Applications. Biology (Basel). 2020, 9, 1–18, doi:10.3390/biology9100336.

2020

Characterization of sterilization methods, problems of sterilization of hydrogels containing nanoparticles

In situ gel; sterilization

AbuhanoÄŸlu, G.; Özer, A.Y. Radiation Sterilization of New Drug Delivery Systems. Interv. Med. Appl. Sci. 2014, 6, 51–60, doi:10.1556/imas.6.2014.2.1.

2014

Some studies on the effect of radiation sterilization on PVA, HEMA, EGDMA, xyloglucan hydrogels are presented

Hydrogel; sterilization

Karami, P.; Stampoultzis, T.; Guo, Y.; Pioletti, D.P. A Guide to Preclinical Evaluation of Hydrogel-Based Devices for Treatment of Cartilage Lesions. Acta Biomater. 2023, 158, 12–31, doi:10.1016/j.actbio.2023.01.015.

2023

Problems of sterilization of hydrogel materials, possible methods of sterilization are presented

Hydrogel; sterilization

Reviewer 5 Report

Comments and Suggestions for Authors

The necessary changes have been made. The article can be published.

Author Response

Comment 1: The necessary changes have been made. The article can be published.

Response 2: Thank you very much for appreciating our manuscript and recommending it for publication in the honorable journal Polymers! 

Round 3

Reviewer 3 Report

Comments and Suggestions for Authors

.

Comments on the Quality of English Language

the authors mentioned that ISS is corrected in the manuscript but still it is not corrected.